# A Biallelic Truncating Variant in the TPR Domain of GEMIN5 Associated with Intellectual Disability and Cerebral Atrophy

**DOI:** 10.3390/genes14030707

**Published:** 2023-03-13

**Authors:** Nazia Ibrahim, Shagufta Naz, Francesca Mattioli, Nicolas Guex, Saima Sharif, Afia Iqbal, Muhammad Ansar, Alexandre Reymond

**Affiliations:** 1Department of Zoology, Lahore College for Women University, Lahore 54000, Pakistan; 2Department of Zoology, University of Sialkot, Sialkot 51040, Pakistan; 3Center for Integrative Genomics, University of Lausanne, 1015 Lausanne, Switzerland; 4Bioinformatics Competence Center, University of Lausanne, 1015 Lausanne, Switzerland; 5Department of Ophthalmology, University of Lausanne, Jules-Gonin Eye Hospital, Fondation Aisle des Aveugles, 1004 Lausanne, Switzerland; 6Advanced Molecular Genetics and Genomics Disease Research and Treatment Centre, Dow University of Health Sciences, Karachi 74200, Pakistan

**Keywords:** TPR domain, neurodevelopmental disorder, cerebral atrophy, autosomal recessive, aspartic acid deletion, consanguinity, intellectual disability

## Abstract

GEMIN5 is a multifunctional RNA-binding protein required for the assembly of survival motor neurons. Several bi-allelic truncating and missense variants in this gene are reported to cause a neurodevelopmental disorder characterized by cerebellar atrophy, intellectual disability (ID), and motor dysfunction. Whole exome sequencing of a Pakistani consanguineous family with three brothers affected by ID, cerebral atrophy, mobility, and speech impairment revealed a novel homozygous 3bp-deletion NM_015465.5:c.3162_3164del that leads to the loss of NM_015465.5 (NP_056280.2):p. (Asp1054_Ala1055delinsGlu) amino acid in one of the α-helixes of the tetratricopeptide repeats of GEMIN5. In silico 3D representations of the GEMIN5 dimerization domain show that this variant likely affects the orientation of the downstream sidechains out of the helix axis, which would affect the packing with neighboring helices. The phenotype of all affected siblings overlaps well with previously reported patients, suggesting that NM_015465.5: c.3162_3164del (NP_056280.2):p. (Asp1054_Ala1055delinsGlu) is a novel *GEMIN5* pathogenic variant. Overall, our data expands the molecular and clinical phenotype of the recently described neurodevelopmental disorder with cerebellar atrophy and motor dysfunction (NEDCAM) syndrome.

## 1. Introduction

Intellectual disability (ID) and development delay (DD) are characterized by limitations in both intellectual functioning and adaptive behavior [1,2]. They affect 1–3% of the world population and represent a significant socio-economic burden on society and the healthcare system [3]. Both genetic and environmental factors can cause ID, but the root causes of approximately 40% of cases are still a mystery. Roughly half of all cases of ID can be traced back to environmental causes. These environmental causes include malnutrition, repeated pregnancies with no gap between them, prenatal/perinatal brain ischemia, postnatal infections, and inadequate medical care, as well as a number of other factors for which an explanation can be found [4]. Additionally, genetic factors, which can have a significant impact, might be the root of limitations in cognitive performance and adaptive behavior. A genetic cause, such as chromosomal abnormalities or gene variants, accounts for the other half of ID cases [5]. In around 2006, research projects involving extremely inbred populations from South East Asia, the Middle East, and North Africa significantly sped up the process of identifying gene mutations that cause ARID [6]. While de novo heterozygous variants are common in the outbred population of western countries [5], autosomal recessive variants are prevalent in the consanguineous population [7]. Although the development of high-throughput sequencing has boosted the identification of pathogenic variants, a large proportion of autosomal recessive intellectual disability (ARID) cases remains unresolved [8].

There are approximately one billion individuals practicing endogamy worldwide, primarily in Muslim countries in the Middle East, Africa, and South Asia. Pakistan, with a rate of 65%, has one of the highest global rates of consanguinity, followed by India (55%), Saudi Arabia (50%), Afghanistan (40%), Iran (30%), Egypt, and Turkey (20%) [9]. Because of the high rate of cousin marriages in Pakistan, genetic disorders like ARID are becoming more common. On average, there are 1.1 cases of severe ID and 6.2 cases of mild ID for every 100 live births [6,10]. Analysis of the large consanguineous families through combined approaches of next-generation sequencing and functional and bioinformatics analyses should allow researchers to understand better the genetic causes of neurodevelopment disorders and efficient interpretation of novel ARID genes.

RNA-binding proteins (RBPs) are important regulators of RNA metabolism and play an essential role in gene expression and regulation [11,12]. GEMIN5 is a predominantly cytoplasmic RBP required for the assembly of the survival motor neurons (SMN) complex, which is involved in the spliceosomal small nuclear RNPs biogenesis [13,14]. It also acts as a signal recognition particle-interacting protein [15], ribosome-interacting factor [16], and mRNA translation regulator [17]. The human *GEMIN5* encodes a 170-kD protein with a specific structural arrangement of its functional domains: the WD40 repeats domain located at the N-terminal region (1-739AA) is involved in the recognition of snRNAs [17,18]. In the middle part (845-1097AA), a tetratricopeptide repeat (TRP) like dimerization domain with 17 helices that oligomerizes as a canoe-shaped homodimer is vital for protein architecture and activity [19], and towards the C-terminus (1287–1508AA), a bipartite non-conventional RNA-binding site (designated as RBS1-RBS2) exerts multiple cellular functions such as RNA binding specificity and affinity [20,21], as well as translation repression and control of the protein stability [22].

Recently, biallelic truncating and missense variants in *GEMIN5* have been linked to a neurodevelopmental disorder characterized by cerebellar atrophy, developmental and cognitive delay, ataxia, motor dysfunction, and hypotonia named NEDCAM syndrome (neurodevelopmental disorder with cerebellar atrophy and motor dysfunction, OMIM#619333) [13,23,24,25]. Here, we report a novel homozygous 3bp-deletion NM_015465.5:c.3162_3164del; (NP_056280.2):p. (Asp1054_Ala1055dlinsGlu) in three affected siblings of a Pakistani consanguineous family. The risk that a newborn will develop ID is increased when both parents are first cousins or have a family history of this disease. It is recommended that families whose children have been diagnosed with ID get genetic counselling because there is a 25% probability of recurrence of ID in future generations.

## 2. Materials and Methods

### 2.1. Molecular Investigation

The current study was approved by the Research Ethics Institutional Review Board (IRB) (Approval number: RERC/LCWU/ZOO-681) of the Lahore College for Women University and the “Commission d’éthique de la recherche” of the canton of Vaud (Protocol number: CER-VD 2021-01400). General physicians, pediatricians, and psychiatrists assessed intellectual and adaptive functioning of affected individuals using standardized assessments with the individual and interviews with family members, teachers, and caregivers. Information related to age, gender, height, weight, head circumference (OFC), and family history was also collected. In addition, magnetic resonance imaging (MRI) and electroencephalograms (EEG) were performed on the two siblings who were affected by the condition, and a radiologist examined the results of the MRI scans. Informed consent forms were obtained from guardians of all affected individuals participating in this study.

DNA was extracted from whole blood samples using a modified version of the LCWU standard non-organic procedure [26,27]. Blood samples were thawed in warm water in preparation for RBC lysis. TE buffer was added (10 mM Tris HCl + 0.1 mM EDTA) and shaken vigorously. Centrifuged at 3000 rpm at 25 °C for 25 to 30 min. To attain a pellet of WBC, three to four washings were given that removed hemoglobin completely. TNE buffer (10 mM Tris HCl + 0.1 mM EDTA + 400 mM NaCl) (6 mL/10 mL of blood), 10% SDS (250 μL/10 mL) and Proteinase K (50 μL 10 mg/mL) were added to WBC pellets to break down proteins. After that, samples were shaken overnight at 37 °C in an incubator. It was then mixed with 1 mL/10 mL of chilled 6 M NaCl and shaken till it became foamy. After 10–15 min at low temperature, Phenol Chloroform Isoamyl alcohol (PCI) (2 mL/10 mL blood) was added. To pellet down the salts and proteins, the mixture was centrifuged at 3000 rpm and 4 °C for 25 min. Centrifuged at 3000 rpm, 25 °C, for 20 min after adding an equal volume of isopropanol to the aqueous solution. To clean the DNA pellet, we used 70% ethanol (5 mL/10 mL of blood). In order to preserve the DNA, a low TE buffer (10 mM Tris HCl + 0.1 mM EDTA) was added to the dried pellet (1.5 mL/10 mL of blood).

WES (whole exome sequencing) was performed on two of the three affected individuals of the family (II.3 and II.4). Genomic DNA was isolated from whole blood. Subsequent library preparation and sequencing were performed at the Lausanne Genomic Technologies Facility. In detail, the exome was captured using the xGen Exome Research Panel v2 (Integrated DNA Technologies, Coralville, IA, USA) and sequenced using the Illumina HiSeq4000 (San Diego, CA, USA) platform according to the manufacturer’s protocols. Sequencing data were analyzed as previously described [28]. Briefly, duplicated reads were marked with Picard Tools (https://broadinstitute.github.io/picard/ accessed on 12 March 2021) and aligned to the hg19 reference genome following the Burrows–Wheeler algorithm [29]. Variants were then called using the Genome Analysis Toolkit (Broad Institute, v4) [30]. The overall mean-depth base coverage was 121- and 137-fold (respectively for individual II.3 and II.4), while on average, 97% of the targeted region was covered at least 20-fold in both patients.

Variants were annotated and filtered using the Varapp software (Swiss Institute of Bioinformatics, v2) [31]. Variants shared by the two affected individuals (II.3 and II.4) and following either a recessive or an X-linked inheritance scenario were analyzed. Homozygous and hemizygous variants passing the quality filter and predicted to have an impact on the protein-coding region with a MAF < 1% (1000g, EVS, EXAC, gnomAD) were analyzed (Appendix A). Variants were then prioritized based on the prediction software tools (CADD, SIFT, PolyPhen2, GERP) and literature. Among the 16 shared rare homozygous variants, only an inframe deletion in *GEMIN5* (Appendix A) is a gene previously associated with an autosomal recessive neurodevelopmental disorder with cerebellar atrophy and motor dysfunction (OMIM #619333). The other 15 variants were excluded, as they are either present at the homozygous state in gnomAD and/or are predicted to be tolerated by different prediction software tools and/or are associated with a different phenotype (Appendix A).

Sanger sequencing was then performed to confirm the presence and correct segregation of the *GEMIN5* variant: NM_015465.5:c.3162_3164del; (NP_056280.2):p. (Asp1054_Ala1055delinsGlu) in the family (Primer Forward: 5′-AGGAACCTCAACCCATGCTGGATT-3′; Primer Reverse: 5′-GCACAGCTTGTGCAACTGGGTTAT-3′).

### 2.2. Protein Model

The 1.95 Angstrom resolution of the human GEMIN5 protein PDB entry 6RNQ [19], which covers residues 845 to 1096 of the dimerization domain, was used to determine the possible impact of the NM_015465.5 (NP_056280.2):p. (Asp1054_Ala1055delinsGlu) variant with Swiss-PdbViewer ( Swiss Institute of Bioinformatics, V4.1) [32].

## 3. Results

### 3.1. Clinical Description

Three affected brothers (out of six siblings) were born from consanguineous healthy parents (Figure 1a). They present with ID, dysarthric speech, and impaired walking (Table 1). In detail, II.2 is a 46-year-old male with 159.5 cm height, 40 kg weight, and 51 cm occipitofrontal circumferences (OFC). He had a delay in sitting, standing, and walking and he currently has an unsteady walk. He has severe ID and speaking impairment. He is also presenting with behavioral anomalies (impaired social skills and aggressiveness). Electroencephalogram (EEG) showed no seizures and magnetic resonance imaging (MRI) scans revealed cerebral atrophy (Figure 1a).

The younger brother II.4 is a 28-year-old man with 167.6 cm in height, 56 kg in weight, and 53.3 cm OFC. He had delayed sitting, standing, and walking during his childhood and still has an unsteady walk. He presents with mild ID, dysarthric speech, and aggressive behavior. EEG report of II.4 reported seizure and MRI scans displayed mild cerebral atrophy along sinus mucosal disease (Figure 1a). The third individual (II.3) is 35 years old with 161.5 cm height, 54 kg weight, and 53.3 cm OFC. He also had delays in sitting, standing, and walking but now can walk. He has a mild ID. Unlike his brothers, he can speak properly, and he does not present any behavioral anomalies.

### 3.2. WES Analysis

WES performed on the DNA of individuals II.3 and II.4 revealed the presence of 16 shared rare homozygous variants (MAF < 1%) (Appendix A). Of these variants, 15 were excluded as they are either present at the homozygous state in gnomAD and predicted to be tolerated by different prediction software tools or associated with a different phenotype (Appendix A). The remaining variant is a homozygous three-base pair deletion NM_015465.5 (NP_056280.2):p. (Asp1054_Ala1055delinsGlu) in *GEMIN5*. According to the American College of Medical Genetics (ACMG) guidelines [33], the variant has two moderate evidence of pathogenicity (PM4: protein length changes due to in-frame deletions/insertions in a non-repeat region and PM2: the variant is absent from controls in gnomAD, Iranome, Regeneron, TopMed, and in our local database of Pakistani individuals) and two supportive evidence of pathogenicity (PP1: co-segregation with the disease in multiple affected family members in a gene definitely known to cause the disease and PP4: patient’s phenotype is specific for the disease). It is thus a likely-pathogenic variant. Furthermore, the segregation analysis in the available relatives validated that the variant segregates with disease status (Appendix A) and follows a recessive inheritance, i.e., it is present at the homozygous state in the three affected brothers while their parents (I.1, I.2) and unaffected sibling (II.6) are heterozygous (Figure 1a and Appendix A). This variant causes the deletion of Aspartate codon 1054 NM_015465.5 (NP_056280.2):p. (Asp1054_Ala1055delinsGlu), a well-conserved amino acid in the tetratricopeptide repeats (TPR) domain of GEMIN5 (PhastCons = 1; PhyloP100 = 7.05; GERP score = 5.76) (Figure 1c). TPRs are dimerization domains that adopt a canoe shape upon folding which is important for protein dimerization and consequently its structure and function [19].

### 3.3. 3D Model

The NM_015465.5 (NP_056280.2):p. (Asp1054_Ala1055delinsGlu) variant is embedded in the middle of the antepenultimate helix of the dimerization domain (Appendix A). The penultimate, antepenultimate, and previous helices participate in the formation of a four-helix bundle with the second helix of the other monomer with residues Lue1059, Lys1062, Ala1066, Ser1067, Thr1070, Glu1073, Lue1074, IIe1077, Val1078, Gly1079 that are near (≤5 angstroms) residues Lys860, His864, Cys867, Lue868, Thr872, His875 of the other monomer (Appendix A). If we postulate that a helix could still be formed upon deletion of Asp1054, the position of downstream residues of this helix would be shifted by one residue (Figure 2a), affecting the orientation of the sidechains out of the helix axis. Four residues of the antepenultimate helix would be changed (Figure 2a). Ala1056 would be replaced by a Lys, with a much longer sidechain that would prevent proper packing of the penultimate and last helix of the dimerization domain through collision with Leu1087 residue. Val1058 would be replaced by a Leu, with a bulkier sidechain that would collide with the sidechain of Ala1042. Leu1059, which packs with His864 and Cys867 of the other monomer, would be replaced by an Ala, whose smaller sidechain will not fill the space as well and perturb the optimal packing. Ala1060 would be replaced by a bulkier Lys sidechain pointing toward the penultimate helix and colliding with Leu1087 (Figure 2a). The deletion of NM_015465.5 (NP_056280.2):p. (Asp1054_Ala1055delinsGlu) will affect the orientation of the sidechains out of the helix axis and disrupt the proper packing of the four-helix bundle located at each end of the dimerization domain (Appendix A). This might not prevent the formation of a dimer, as a significant interaction surface could be formed, but disruption of the three last helices would affect the proper fold position of the C-terminal region relative to the dimerization domain.

## 4. Discussion

Bi-allelic truncating and missense variants in *GEMIN5* have been associated with NEDCAM syndrome, a neurodevelopmental condition characterized by cerebellar atrophy, ID, and motor dysfunction [13,23,24,25]. ID is defined as having an IQ of less than 70 and deficits in adaptive behavior or daily living skills (eating, dressing, communication, participation in group activity). Individuals with intellectual disabilities learn slowly and struggle with abstract concepts [34]. Here, we report a homozygous 3bp-deletion variant in *GEMIN5* in three siblings born from consanguineous parents presenting with ID, cerebral atrophy, walking and speaking impairment, and motor dysfunctions (Table 1). The variant identified in our study would lead to the loss of NM_015465.5 (NP_056280.2):p. (Asp1054_Ala1055delinsGlu), an evolutionarily conserved amino acid located in the TPR domain of GEMIN5 (Figure 1c). According to our in silico analysis, the deletion of this amino acid would impact the helix conformation in the dimerization domain and consequently its function.

Overall, 45 affected individuals, including our patients, have been reported [13,23,24,25] (Table 1). All of the patients presented with cerebellar atrophy and intellectual disability, with the exception of one who was first diagnosed with a moderate form of the condition but was later deemed to be normal [25], showing that these are the main hallmarks of NEDCAM syndrome (Table 1 and Appendix A). Other recurrent features are developmental (*n* = 43/45), motor (*n* = 41/43), cognitive (*n* = 37/42), speech delay (*n* = 39/41), ataxia (*n* = 31/42), and walking issues (*n* = 40/42) (Table 1 and Appendix A) [13,23,24,25]. The wide clinical spectrum and variable severity of the disease were suggested to result from differences in expression levels of GEMIN5 than previous protein expression analyses have shown [13,24,25].

Many NEDCAM syndrome missense variants (Ala994Val, Arg1014Gln, His913Arg, Ala1007Thr, Lue1068Pro, His923Pro, and Ser1000Pro) [24] and one deletion (Arg899Pro fs*) are located in the linker-dimerization domain of GEMIN5 which controls protein–protein/RNA interactions and dimerization by connecting the WD and RBS domains [19,35,36] (Figure 2b). According to our 3D model, the variant Arg899Pro fs*, located in a helix at the start of the dimerization domain, would lead to the premature termination of this helix and prevent dimerization. The 3D model suggests that Ser1000Pro points toward the solvent and are highly exposed, while His913Arg, His923Pro, Ala1007Thr, Leu1068Pro, and Ala994Val are sandwiched between two helices of the same monomer of protein. His923Pro and Leu1068Pro similar to Arg899Pro fs* would lead to premature helix termination (Figure 2b). Consistent with this hypothesis, functional analysis of the homozygous Leu1068Pro and His923Arg in human patient-derived iPSC neurons showed a significant reduction of the snRNP complex proteins, suggesting a disruption of the dimerization ability of this protein interferes with its key functions [19,24]. Arg1014Q is in the dimerization interface and Arg1016Cys, located in a solvent-facing loop between two helices, has oxidoreduction potential (Figure 2b) [23]. Asp1019Cys and Arg1016Cys have been reported to disrupt the dimerization properties of GEMIN5, indicating their possible role in protein–protein interactions [13,23]. Our NM_015465.5 (NP_056280.2):p. (Asp1054_Ala1055delinsGlu) variant will not directly affect the snRNA binding domain located in the WD40 repeat N-terminal region, nor the ribosomal binding sites located in the C-terminal la region. NM_015465.5 (NP_056280.2):p. (Asp1054_Ala1055delinsGlu) is predicted to affect the packing with other helices of the TPR dimerization domain by modifying the orientation of sidechains of the antepenultimate helix. The TPR domain is immediately followed by a C-terminal region (residues 1097–1508), which consists of an unstructured region and RNA-binding site. Region 1287–1508 has been reported to repress translation [22], and disruption of the proper dimerization of the TPR domain has been further demonstrated to affect translation stimulation [19]. Overall, the variant reported in this study confirms the deleterious effect of GEMIN5 TPR variants on its structural conformation, interaction, and function.

As our three patients have a homozygous three-base pair deletion in the TPR domain (Figure 1a), we further evaluated if individuals carrying a variant in this domain present with specific clinical features compared to patients with a variant in a different region [23,24] (Table 1 and Appendix A). Upon comparison of the 17 individuals with variants in the TPR domain with the 15 individuals with a variant in a different region [13,23,24,25] (Table 1 and Appendix A), we did not observe any specific clinical features in patients with variants in the TPR domain (Table 1), suggesting that NEDCAM variants alter the function of GEMIN5 in the same way independently of their localization.

In conclusions, the consistency of our patients’ core features with those of other, previously-reported *GEMIN5* patients and pathogenicity detected by different in silico prediction tools collectively suggest that NM_015465.5:c.3162_3164del; NP_056280.2: p.Asp1054_Ala1055delinsGlu) is a novel cause of underlying autosomal neurodevelopment disease characterized by cerebral atrophy and intellectual disability. Proper genetic counseling for the affected family is essential in the case of rare genetic diseases. Eventually, it will help in reducing the total burden of the disease on the population. Furthermore, parenteral genetic screening/diagnosis is the best strategy for managing this disease, which currently has no therapy.

## Figures and Tables

**Figure 1 genes-14-00707-f001:**
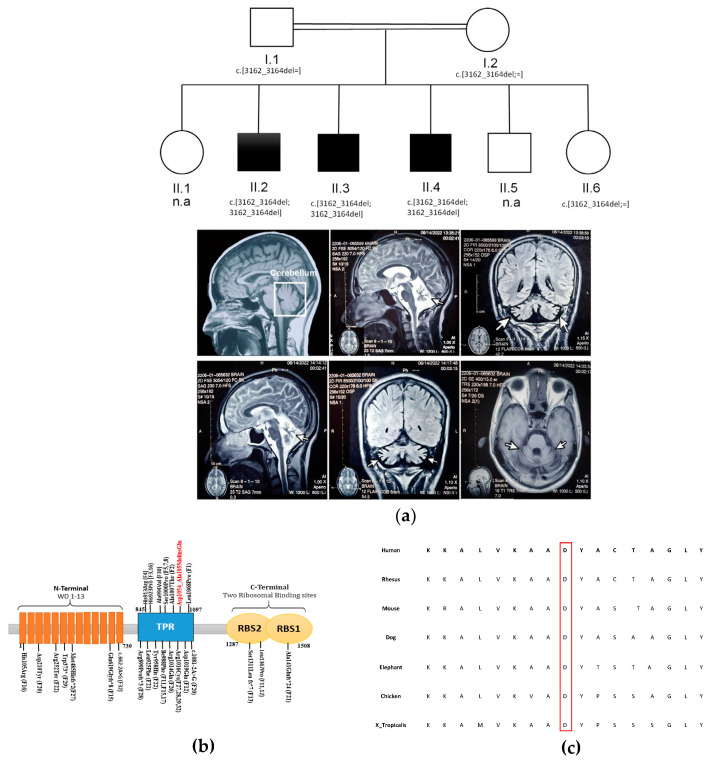
NM_015465.5 (NP_056280.2):p. (Asp1054_Ala1055delinsGlu) deletion in *GEMIN5* in three siblings. (**a**) On the top is the family pedigree with the 3bp-deletion variant genotype of individuals. On the bottom, brain MRI of normal individual (on the top left sagittal view), II.2 (on the top center sagittal view and on the top right frontal) and II.4 (on the bottom left sagittal view, in the bottom center the frontal, and on the bottom right the axial view) showing characteristic cerebral atrophy (white arrows) in patients carrying three base pair deletion NM_015465.5:c.3162_3164del; NP_056280.2: p.Asp1054_Ala1055delinsGlu) in *GEMIN5*. (**b**) Schematic representing functional domains and position of NM_015465.5 (NP_056280.2):p. (Asp1054_Ala1055delinsGlu) (highlighted as red) and previously reported biallelic variants (highlighted as black). Family numbering is given for previously reported variants. (**c**) Alignments of protein sequence across different species representing high conservation of the deleted NM_015465.5 (NP_056280.2):p. (Asp1054_Ala1055delinsGlu) identified in the three brothers, highlighted in the red rectangle.

**Figure 2 genes-14-00707-f002:**
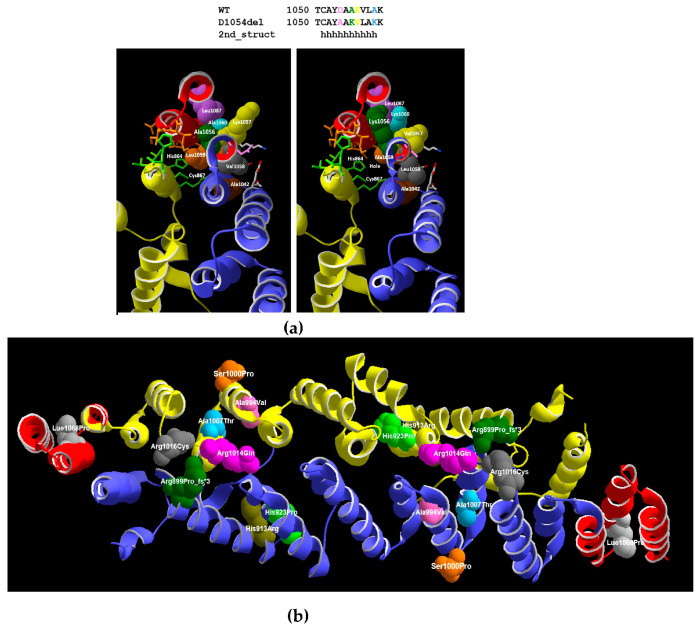
3D-protein modeling of *GEMIN5* variants. (**a**) The predicted effect of the NM_015465.5 (NP_056280.2):p. (Asp1054_Ala1055delinsGlu) variant. On the left is the wild-type protein, and on the right is the mutated one. NM_015465.5 (NP_056280.2):p. (Asp1054_Ala1055delinsGlu) is indicated in pink in the helix. Every amino acid is presented with a specific color as follows: Ala1056 (dark green space fill) Leu1087 (lavender space fill) Val1058 (dark grey space fill) A1042 (brown space fill). Leu1059 (orange space fill), Ala1060 (cyan space fill) to show its relative position in a helix on the left and the right side; (**b**) Previously reported variants in the TPR domain of *GEMIN5* presented with different colors: His913Arg (olive), His923Pro (green), Ser1000Pro (orange), Ala1007Thr (sky blue), Leu1068Pro (silver), Arg899Pro_fs*3 (dark green), Ala994Val (pink), Arg1014Gln (magenta) and R1016Cys (nickel).

**Table 1 genes-14-00707-t001:** Clinical features of patients with biallelic *GEMIN5* variants.

	Phenotypes of Patients with Biallelic *GEMIN5* Variants [13,23,24,25] and Our Study	Our Study			Phenotypes of Patients with Variants in the TPR Domains [13,23,24] and Our Study	Phenotypes of Patients with Variants Outside the TPR Domains [13,23,24,25]
Total Patients	45	II.2	II.3	II.4	17	15
Sex; male/female (M/F)	69% M (31) 31% F (14)	M	M	M	59% (10) M, 41% (7) F	73% (11) M, 27% (4) F
Onset		<1	After 5 years	<1		
Age at last evaluation		46 Y	35 Y	28 Y		
Origin		Pakistani	Pakistani	Pakistani		
Language		Punjabi	Punjabi	Punjabi		
Consanguineous parents (Number of families for yes)	6/28 (21%)	+	+	+	5/10 (50%)	0/7 (0%)
Genetic testing						
Kind of Variant		Homozygous inframe deletion	Homozygous inframe deletion	Homozygous inframe deletion		
Chromosome number: Transcript		Ch5: (NM_015465.5; NP_056280.2)	Ch5: (NM_015465.5; NP_056280.2)	Ch5: (NM_015465.5; NP_056280.2)		
Variant (g. DNA; protein)		c.3162_3164del; p. (Asp1054_Ala1055delinsGlu)	c.3162_3164del; p. (Asp1054_Ala1055delinsGlu)	c.3162_3164de; p. (Asp1054_Ala1055delinsGlu)		
Growth at last investigation						
Height (cm)		159.6	161.5	167.6		
Weight (kg)		40	54	56		
Head circumference (cm)		51	53.3	53.3		
Development						
Delayed	96% (43/45)	+	+	+	94% (16/17)	93% (14/15)
Cognitive delay	88% (37/42)	+	+	+	100% (15/15)	73% (11/15)
Intellectual disability (Mild to severe)	98% (41/42)	Severe	Mild	Mild	100% (15/15)	93% (14/15)
Motor delay	95% (41/43)	+	-	+	88% (15/17)	94% (12/13)
Speech delay	95% (39/41)	+	-	+	93% (13/14)	92% (14/14)
Regression (−)	100% (42/42)	-	-	-	100% (16/16)	100% (13/13)
Neurological finding						
Ataxia	74% (31/42)	+	-	+	60% (9/15)	79% (11/14)
Walking difficulties	95% (40/42)	+	-	+	87% (13/15)	100% (14/14)
Neurological Evaluation						
MRI Scans	100% (44/44)	Diffused Cerebral atrophy	Not done	Mild diffuse cerebral atrophy	100% (16/16)	100% (15/15)
Behavioral anomalies						
Aggressive behavior		+	-	+		
Seizures		-		+		
Clinical Course						
Progressive/static (P/S)	78% S (29/37) 22% P (8/37)	P	S	S	86% (12) S, 14% (2) P	83% (10) S, 17% (2) P
Others		Sinus mucosal disease, no maleness		Sinus mucosal disease		

## Data Availability

Publically no dataset created and it is not available.

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
