# Peer review of "A Biallelic Truncating Variant in the TPR Domain of GEMIN5 Associated with Intellectual Disability and Cerebral Atrophy"

_genes, 2023, doi:10.3390/genes14030707_

Round 1

Reviewer 1 Report

Paper for review

Nazia Ibrahim et al reported a patient having homozygous variant in the GEMIN5 that was identified using WES. The patient revealed intellectual disability and cerebral atrophy. Previous;y already 33 mutations have been reported in the GEMIN5 gene associated with NDD.

The clinical and molecular data very are very interesting and could be of interest to the readers. However, there are some comments that need to be addressed.

General comments:

+ Follow nomenclature: https://varnomen.hgvs.org/

*- The reference sequence (NP_) and NM_ should also be added.

**Gene name should be in italics.

USE (OMIM #), and use the proper name of the disorder.

Use the term variant……… mutation is not correct term.

Extensive English language editing required.

Methods/Results

WES protocol and filtration steps should be explained. How many variants were shortlisted and subjected to Sanger sequencing?

Was Sanger sequencing performed? The author did not mention Sanger seq.

Results

It would be better to add the age, height, weight, and HC of the patients with SD.

Kindly check the mutation position:

Looks incorrect: GEMIN5 (NM 015465.5, c.3162 3164delAAA, p.Asp1054del)

Correct: GEMIN5 (NM_015465.5):c.3162_3164del; (p.Asp1054_Ala1055delinsGlu)

How the variants were classified as pathogenic, likely pathogenic, VUS? ACMG criteria fulfilled?

The variants look like VUS; why the author said it's pathogenic?

+++Why are Sanger electrograms not shown? Kindly add to figure 1.

**Was the patient treated with any medication? Condition of patients before and after treatment?

3D Model: Any energy difference obtained between the wild type and mutated protein? Any effect on the active site? Binding? Downstream signaling?

Table 1: No need to add Cast and occupation.

Why consanguinity is not mentioned for the other studies? Add %.

Ada a table showing all the mutations reported to date.

Discussion

Any genotype-phenotype correlation? Location of variants associated with variable phenotypes?

·         In the last discussion add lines for future perspectives; discuss newborn screening, NIPT, PGT-A, PGT-M, for example.: Proper genetic counseling for the affected family is essential in the case of rare genetic diseases. Furthermore, parenteral genetic screening/diagnosis is the best strategy for managing this disease, which currently has no therapy (PMID: 31557427; PMID: 33613643; PMID: 3380482; PMID: 36406136)

Figure:

Figure 1: part a: black strips should be added to the patient’s eyes. Alternatively, it should be mentioned in the methods that consent for facial photo publishing was obtained. Part c? The mutated a.a sequence is missing. Kindly add.

References that would be added, if necessary, from the Pakistani population:

PMID: 36527195, PMID: 36190665, PMID: 36604604, PMID: 35782384, PMID: 34702355.

Author Response

Dear Reviewer,

Thank you for your valuable suggestions, point by point responses have been provided in attached word file

Reviewer 2 Report

A bialletic truncating variant in the TPR domain of GEMIN5 2 associated with intellectual disability and cerebral atrophy

Ibrahim et al. has identified a novel 3bp deletion in GEMIN5 genes that is associated with intellectual disability (ID) in the Pakistani family. They have also provided a 3D structure of the wild type and deleted structure showing the difference in the folding of the protein on deletion. The novel variant would be helpful for screening of the patient affected with ID. 

However I have some concerns regarding the methodology and result of the manuscript. Authors should make particular changes which as follows:

Major Revision

  1. The author has written a very brief method section. I would advise in detail so that readers could understand what they have done and how they have done. Please write it what are the values were taken for eg. For variant filtering, what is the global allele frequency cutoff has been considered. 

  2. When I searched for a PDB entry for GEMIN5 https://www.rcsb.org/structure/unreleased/5RNQ. I could not find the result. Please provide the link.

  3. In Figure 1, Please remove the photos of the patients. It is not ethically correct and also there is no need to put it so please try to hide the identity of the patients as much as possible.

  4. How does the author know that Parents and unaffected siblings of the patients are heterozygous? If they have done Sanger sequencing of parent and unaffected siblings, please provide the Sanger result in the supplementary.  

  5. Also please provide the Integrated Genome Viewer result in the supplementary for Patient II.3 and II.4 for whom exome sequencing were performed.

  6. I understand that the region is highly conserved. But why does the CADD score low?

Minor Revision

  1. In the title of manuscript please change from “bialletic” to “biallelic”

  2. Page 1 line 39-40 Please rewrite this statement “About one billion individuals worldwide live in countries practicing endogamy” to “There are approximately one billion individuals practicing endogamy worldwide with 65% of Pakistani populations following consanguineous marriage”.

  3. Please correct the references 23, 25.

  4. For the Varapp software please provide some information other than variant calling. How does it perform the variant filtering and variant annotation?

  5. In Table 1, it might be confusing for the reader “45, 17, 15” represents? Please change the row name from Individual to Individual number or total patients.

  6. In WES analysis line 107, 108. It cannot be both and or. From the data it looks like it is or. Please put it like that.

  7. In Fig1a after removing the patient the photo author can put the normal brain MRI so the reader can correlate with it.

Author Response

Dear Reviewer,

Many thanks for your good suggestions, point by point responses have been provided in attached in word file.

Round 2

Reviewer 2 Report

I appreciate the authors for the correction. Still I have some comments that needs to be resolved. The following are:

1. I would advise the author to classify the disease variant based on the  American College of Medical Genetics and Genomics (ACMG) and Associate of Molecular pathology (ACMG-AMP) classification into pathogenic, likely pathogenic, VUS, Benign, Likely benign.

2. I could not find Supplementary Figure 1, may be authors forgot to upload it.

3. In Fig1a as I have earlier suggested author can put the normal brain MRI figure so that author can correlate. it is not done. Please do it so that reader can understand.

Author Response

Response to Reviewer 2 Comments

Point 1: I would advise the author to classify the disease variant based on the  American College of Medical Genetics and Genomics (ACMG) and Associate of Molecular pathology (ACMG-AMP) classification into pathogenic, likely pathogenic, VUS, Benign, Likely benign

Response 1: Thank you for your valuable suggestion. We added explanation in WES results part.

In detail; According to the ACMG guidelines (Richards et al 2015), the variant has two moderate evidence of pathogenicity (PM4: protein length changes due to in-frame deletions/insertions in a non-repeat region and PM2: the variant is absent from controls in gnomAD, Iranome, Regeneron, TopMed and in our local database of Pakistani individuals) and two supportive evidence of pathogenicity (PP1: co-segregation with the disease in multiple affected family members in a gene definitely known to cause the disease and PP4: patient’s phenotype is specific for the disease). It is thus a likely-pathogenic variant.

Point 2: I could not find Supplementary Figure 1, may be authors forgot to upload it.

Response 2: Yes, I forgot to upload. Kindly find it now in attachment

Point 3: In Fig1a as I have earlier suggested author can put the normal brain MRI figure so that author can correlate. it is not done. Please do it so that reader can understand

Response 3: Yes, it’s a good suggestion. Figure revised by adding brain MRI scan of normal person. The cerebellum part is highlighted so that reader understand which area is affected in cerebral atrophy condition.
